# The Role of miRNA for the Treatment of MGMT Unmethylated Glioblastoma Multiforme

**DOI:** 10.3390/cancers12051099

**Published:** 2020-04-28

**Authors:** Anna Kirstein, Thomas E. Schmid, Stephanie E. Combs

**Affiliations:** 1Institute of Radiation Medicine (IRM), Department of Radiation Sciences (DRS), Helmholtz Zentrum München, 85764 Neuherberg, Germany; 2Department of Radiation Oncology, Technical University of Munich (TUM), Klinikum Rechts der Isar, 81675 Munich, Germany; 3Deutsches Konsortium für Translationale Krebsforschung (DKTK), Partner Site Munich, 81675 Munich, Germany

**Keywords:** glioblastoma, miRNA, MGMT, survival, radiotherapy, chemotherapy, temozolomide, translational medicine

## Abstract

Glioblastoma multiforme (GBM) is the most common high-grade intracranial tumor in adults. It is characterized by uncontrolled proliferation, diffuse infiltration due to high invasive and migratory capacities, as well as intense resistance to chemo- and radiotherapy. With a five-year survival of less than 3% and an average survival rate of 12 months after diagnosis, GBM has become a focus of current research to urgently develop new therapeutic approaches in order to prolong survival of GBM patients. The methylation status of the promoter region of the O^6^-methylguanine–DNA methyltransferase (MGMT) is nowadays routinely analyzed since a methylated promoter region is beneficial for an effective response to temozolomide-based chemotherapy. Furthermore, several miRNAs were identified regulating MGMT expression, apart from promoter methylation, by degrading MGMT mRNA before protein translation. These miRNAs could be a promising innovative treatment approach to enhance Temozolomide (TMZ) sensitivity in MGMT unmethylated patients and to increase progression-free survival as well as long-term survival. In this review, the relevant miRNAs are systematically reviewed.

## 1. Introduction

Cancer is one of the leading causes of death worldwide with 14 million new cases diagnosed and eight million deaths every year. With 256,000 cases per year, tumors of the central nervous system account for 2% of all diagnosed cancers and are therefore the 17th most common cancer [1]. Each year the American Cancer Society compiles a cancer statistic for the USA, estimating the annual cancer incidences and mortalities based upon mortality data from 1930 to 2017 and incidence data from 1975 to 2016 [2]. For 2020, they estimate 23,890 new brain and other nervous system cases with 18,020 deaths [2]. Although the yearly incidence rate with 4-8 cases per 100,000 worldwide is relatively low, mortality rates are significantly high, making it the 12th most frequent cause of cancer-related deaths. The most common primary malignancy of the central nervous system is Glioblastoma multiforme (GBM) [1].

Glioblastoma multiforme is one of the most common and most aggressive primary brain tumors with a five-year survival of less than 3% [3] and an increasing incidence rate [4]. In recent years, the mechanisms explaining the radio- and chemoresistance of glioblastoma have been extensively studied but are still poorly understood. Radiotherapy with concomitant or adjuvant temozolomide-based chemotherapy following surgery has become the standard treatment for GBM [5]. However, short median survival is still observed in patients with an unmethylated promoter region of the O^6^-methylguanine–DNA methyltransferase (MGMT) [6]. Despite the minor benefit of additional Temozolomide (TMZ) to unmethylated patients and regardless of the treatment regimen, MGMT promoter methylation status is routinely investigated in all patients after surgery as an independent prognostic biomarker. Therefore, an unmethylated MGMT promoter is an inherent prognostic indicator for poor overall-survival, which demonstrates the urgent need for the identification of new prognostic factors, especially for these patients. For this specific patient group, tailored study concepts have been performed with intensified TMZ or with concepts omitting TMZ but adding novel potentially effective substances such as Vascular Endothelial Growth Factor (VEGF)-inhibitors, integrin-antagonists or other molecular targeted substances. To date, all of these studies were negative and have not offered additional benefit [7,8,9,10].

In recent years, circulating microRNAs have been extensively studied as tumor biomarkers to predict therapy outcome and to follow up therapy response. miRNAs are endogenous, single-stranded, non-coding small RNA molecules with a length of about 22 nucleotides [11]. The interaction of miRNAs with the target mRNA leads to the degradation or translational repression of the target mRNA, which ultimately results in the down-regulation of the designated protein. This regulatory network of miRNAs affects many different biological functions and therefore represents a great potential for clinical applications [12]. The up- or down-regulation of miRNAs in tumor cells is deterministic for either a tumor-suppressive or an oncogenic characteristic of the respective miRNA [12].

In this manuscript, the relevant literature investigating the relationship between different miRNAs and glioblastoma was systematically reviewed and the results were analyzed to evaluate the value of different miRNAs in the treatment of GBM.

## 2. Glioblastoma Multiforme

The 2007 World Health Organization (WHO) classification of tumors of the central nervous system [13] is mainly based on microscopic analyses of hematoxylin and eosin-stained sections, immunohistochemistry of lineage-associated proteins and characterization of ultrastructures. Important characteristics include nuclear atypia, mitotic activity, vascularization, necrosis, pleomorphism and microvascular proliferation [14]. In the updated version from 2016 [15], molecular markers are taken into account proposing a more detailed classification of glioblastoma. GBM is classified as grade IV diffuse astrocytic tumor and is characterized by uncontrolled cellular proliferation, diffuse infiltration and intense resistance to radiotherapy [3]. Due to increasing evidence towards a different origin of primary and secondary GBM [16,17,18,19], GBM is now subdivided into isocitrate dehydrogenase (IDH)-wildtype, IDH-mutant and not otherwise specified (NOS) glioblastoma. NOS glioblastoma are either primary or secondary glioblastoma, but a full evaluation of the IDH status is either inconclusive or not performed due to the patient’s age [15]. IDH-wildtype or primary glioblastoma develops rapidly within 3-6 months directly from glial progenitor cells and is characterized by diffuse infiltration, extensive necrosis and a unique mutation pattern [1]. EGFR amplification [17], PTEN mutation [18,20,21] and loss of chromosome 10 [17,18,20] are particular features of primary GBM, as well as the older age of the patients. The median age at diagnosis is 62 years with a male-to-female ratio of 1.46 [14,20], and the median overall-survival is 15 months [15]. A total of 90% of all glioblastoma are primary glioblastoma [14,15]. IDH-mutant or secondary glioblastomas, in contrast, develop over several years from low-grade astrocytomas (WHO grade II) and anaplastic astrocytomas (WHO grade III) and feature a different, unique mutation pattern, which was postulated to be the result of a sequential acquisition [1,19,22,23]. This pattern includes a TP53 mutation [1,17,20], LOH on chromosomes 10q and 19q [1,17,18,20,24] as well as deletion of *p16* [20,25] and inactivation of RB [20,23,25]. Median age at diagnosis is 44 years with a median overall-survival of 31 months and a male-to-female ratio of 1.12 [14,15,19,20]. Although there is no universally accepted glioblastoma stem cell marker and there might be several stem cell markers [26], CD133 expression is significantly higher in primary, compared to secondary glioblastoma [27]. This might explain the intense resistance to chemo- and radiotherapy of primary glioblastoma due to the presence of potential glioblastoma stem cells.

### 2.1. Current Treatment of GBM

Treatment of patients with GBM is always interdisciplinary. For all treatments, the strongest prognostic factors are patient’s age, performance score, tumor volume as well as molecular characterization. Imaging information from magnetic resonance imaging (MRI), computer tomography (CT), positron-emission tomography (PET) as well as other functional imaging, such as 5ALA, provide a basis for solid characterization of tumor extension. After imaging diagnosis, surgical resection of the tumor mass is crucial to relieve symptoms such as headache, vision and memory problems as well as nausea [28] and should be performed following the rules of maximal-safe resection. Resection allows for pathological examinations to confirm the diagnosis and to investigate several molecular markers, such as MGMT and IDH status. The diffuse infiltrative characteristic, as well as extensive vascularization into the surrounding healthy tissue, limits the complete resection of GBM and makes recurrence highly possible [3]. Hence, complete surgical resection is almost impossible and, therefore, surgery is followed by radiotherapy, generally concomitant with chemotherapy to eliminate tumor cells in the microenvironment as well.

In the 1970s, BCNU (bis-chloroethylnitrosourea—carmustine) was discovered and since then administered as an alkylating antineoplastic agent as it was shown to penetrate the blood brain barrier (BBB) and to be effective in treating intracranial neoplasms [29]. However, the combination of BCNU and radiotherapy did not significantly enhance median survival [29].

Since 2005, administration of the oral alkylating agent temozolomide (TMZ) presents the standard agent for GBM patients, as it causes only mild side-effects and efficacy has been proven in clinical trials [5]. It is given as a daily dose of 75 mg per m² body-surface area for five consecutive days for six weeks [28]. After four weeks, the dose is increased to 150 mg per m². Adjuvant, conventional radiotherapy is given in 30 fractions at 2 Gy to a total dose of 60 Gy over a period of six weeks [28]. Alternatively, hyperfractionated radiotherapy is given for 15 days with a total dose of 34 Gy in 3.4 Gy fractions or in 15 daily fractions to a total dose of 10 Gy in 2.6 Gy fractions [28].

After radiochemotherapy with TMZ was introduced, it has been shown that patients with an unmethylated MGMT promoter as well as older patients benefit less from TMZ [30]. However, it has also been shown that even in elderly patients treated with short course radiotherapy concomitant treatment improves outcome [31]. These inconclusive data argue for more accurate discrimination of patient subgroups. A 4-miRNA signature consisting of let-7b-5p, miR-125a-5p, miR-615-5p and let-7a-5p was proposed to assign patients into high- and low-risk groups [32]. Three of the four miRNAs—let-7b-5p, let-7a-5p and miR-125a-5p—are tumor suppressive in GBM and are higher expressed in the low-risk GBM group [32]. Only miR-615-5p does not show a tendency towards a certain expression level in either risk group [32]. This leads to the promising conclusion that this 4-miRNA signature is associated with overall survival of GBM patients. This 4-miRNA could be used to differentiate GBM patients and predict therapy outcome. Still, all possibilities should be evaluated in newly diagnosed as well as recurrent patients, including surgery, radiotherapy and chemotherapy. Again, the extent of surgical resection is crucial [33] and the benefit of radiotherapy for recurrent GBM is evident for resected as well as unresected lesions [34,35,36,37].

Recurrence or progression is almost inevitable and is postulated after a median time of 32 to 36 weeks after treatment completion and a final mortality rate close to 100% [38]. This alone describes the urgent need for treatment improvement and the discovery of alternative treatment regimes.

### 2.2. TMZ and MGMT

Since 2005, the standard treatment of glioblastoma involves early adjuvant chemotherapy with the administration of TMZ [5,39]. TMZ is a prodrug from an imidazotetrazine derivative, which is stable in acidic pH and rapidly hydrolyzes by passing through neutral to basic pH [40,41,42]. Therefore, it survives the gastric acid enabling an oral administration. Due to the lipophilic character of the prodrug, it is able to penetrate the BBB [41]. Only in the brain, where the pH is around 7, spontaneous ring-opening hydrolysis of the imidazotetrazine leads to the formation of the active alkylating metabolite 3-methyl-(triazen-1-yl) imidazole-4-carboximide (MTIC) intermediate [40]. MTIC, in turn, is unstable at pH values below 7 but stable in an alkaline environment [42]. Further hydrolysis of MTIC forms 5-amino-imidazole-4-carboxamide (AIC) and methyl diazonium ions, which react with nucleophilic sites on the DNA producing methyl adducts [41]. There are several sites for DNA methylation, such as N^7^ (70%) and O^6^ (5%) of the base guanine as well as the N^3^ (9%) site of adenine [41,43]. However, only the relatively rare site of the O^6^ position at the base guanine is of importance for the anti-cancer activity of TMZ [41,42] and this site is, therefore, speculated to be mutagenic and cytotoxic [44,45].

During DNA replication, O^6^-methylguanine pairs with thymine creating a wobble base pair. This mismatch is repaired by the DNA mismatch repair (MMR) pathway, which involves the recognition of the mismatch via several mismatch recognition complexes [46]. Single-stranded DNA nicks are created in close proximity to the wobble base pair allowing accessibility to the mismatched base thymine, which is digested by the 5′-3′ exonuclease I [46]. Eventually, DNA polymerase δ fills the gap with a new thymine [46,47]. Continuous rounds of thymine deletion and insertion eventually lead to a depletion of deoxythymidine triphosphates (dTTP). Lack of dTTP will result in a lack of DNA synthesis and ultimately causes cell death via DNA double-strand breaks [47].

MGMT or sometimes also called the O^6^-alkylguanine-DNA-alkyltransferase is a nuclear protein involved in this mismatch repair pathway [48]. MGMT, therefore, protects not only normal cells from apoptosis but also tumor cells. It removes alkyl groups, preferably methyl groups, from the O^6^-methylguanine to counteract the futile circles of thymine deletion and insertion [49]. The removed methyl groups are covalently transferred to a cysteine acceptor residue contained within the active site of MGMT [50]. This results in a conformational change, which leads to degradation of the MGMT protein. As the cysteine site is not regenerated, the reaction is a suicide reaction [50], which makes MGMT a protein and not an enzyme [49]. Hence, the amount of methyl groups that can be removed is limited to the amount of MGMT present in the cell, which is dependent on the MGMT promoter methylation status. So, the absence or presence of MGMT mainly contributes to the chemoresistant character of GBM [48,49].

MGMT, therefore, counteracts the therapeutic efficacy of TMZ and promotes treatment failure. Stupp et al. discovered in their studies from 2000 to 2002 that administration of TMZ starting early in the treatment course and adjuvant to radiotherapy increases median survival to 2.5 months and a resulting survival rate of 27% [5]. This constant treatment regime makes dose escalation possible as well as depletion of MGMT.

In 2005, Hegi et al. published that the promoter methylation status of MGMT is an important prognostic biomarker to predict the TMZ chemotherapy outcome [39]. Overall survival of patients with a methylated MGMT promoter who received radiotherapy plus temozolomide was significantly increased compared to patients with an unmethylated MGMT promoter [39]. MGMT promoter unmethylated patients have no or only little benefit from TMZ adjuvant to radiotherapy, which suggests that other mechanisms play a role to overcome TMZ resistance. Since then, the MGMT promoter methylation status in GBM patients is routinely investigated after surgery to predict which patients would benefit most from TMZ.

Recent studies have shown that MGMT expression does not always correlate with MGMT promoter status and that some individual patients with an unmethylated MGMT promoter show comparable long-term survival [51]. This leads to the assumption that other mechanisms are active in regulating MGMT expression, which includes miRNAs [52]. Therefore, new innovative and personalized treatment options need to be developed, especially for patients with an unmethylated MGMT promoter. Some compounds were already tested or are currently tested in clinical trials for the treatment of unmethylated patients.

### 2.3. Current Diagnostic and Prognostic Biomarkers for GBM

The most commonly analyzed biomarkers in GBM are currently IDH status, MGMT status, 1p/19q co-deletion and ATRX loss [53]. There are, however, several classes of molecules, proposed to aim as biomarkers for GBM detection, which are found in the blood, cerebrospinal fluid (CSF) and urine.

Proteins are detectable in all kinds of body fluids and can be easily withdrawn from the patient. GBM-specific protein markers include VEGF, angiogenesis-associated proteins, extracellular matrix proteins, matrix metalloproteinases, cell line associated proteins, macrophage migration inhibitory factor (MIF) as well as functionally-related proteins, such as CD44 [53,54]. CD44 was shown as a potential marker for survival outcome and treatment resistance [54]. All these have shown deviating amounts and compositions in patients where tumor progression was observed [53].

Another class used for biomarkers are small molecules, such as lipids and metabolites. Due to their low specificity and small size, they can only be used to verify a diagnosis after other markers were tested positive [53].

Circulating tumor cells (CTCs), which are primary tumor cells circulating in the body via the blood stream, for example, might be important in other cancers apart from GBM [53]. As GBM rarely metastasizes and is described as a cranial-restricted tumor, CTCs might not be found in GBM patients in blood samples [53].

Extracellular vesicles, secreted by the tumor and containing material characteristic of the parental cells, can be found in the serum as well as the CSF. It is known, that GBM secrete exosomes, microvesicles, apoptotic bodies and oncosomes containing the glioma-specific receptor of epidermal growth factor (EGFRvIII), miR-21 as well as mutant IDH1 mRNA [53].

Circulating miRNAs have recently gained attention in research and present promising new biomarkers [55]. They can usually be found in peripheral blood of GBM patients and plasma levels of some miRNAs were already shown to be altered [56]. Some of these circulating miRNAs seem predictive in early diagnosis and helpful during treatment monitoring [55].

### 2.4. Innovative Treatment Options for MGMT Unmethylated Patients

Apart from TMZ, other compounds and therapeutic candidates have also been discovered and are currently tested for the treatment of unmethylated patients. Most of these compounds aim for radiosensitization [10,57,58] affecting the DNA repair pathway or other related pathways. However, in the following, two therapy alternatives will be presented, which target MGMT for radiosensitization.

#### 2.4.1. O^6^-Benzylguanine

O^6^-benzylguanine is a guanine analog with antineoplastic activity and has been proposed to serve as a therapeutic agent to improve efficiency of alkylating agents [59]. Since benzyl groups get displaced faster compared to methyl groups, O^6^-benzylguanine would serve as an effective agent to inactivate MGMT [60]. O^6^-benzylguanine binds to the active site of MGMT, thereby transferring the benzyl moiety to the cysteine residue blocking the active site for methyl groups [47]. Dolan et al. have shown that O^6^-benzylguanine enhances the cytotoxicity of alkylating agents, which specifically produce O^6^-methylguanine [61]. They observed a direct correlation in vitro between increased effectiveness of methylating agents upon O^6^-benzylguanine addition and depletion of MGMT [61]. Furthermore, Dolan et al. have shown in vivo that already low doses of O^6^-benzylguanine completely deplete MGMT activity [60]. However, to achieve long-lasting efficiencies, higher doses were required, which exhibited increased acute cytotoxicities, especially to the hematopoietic system. The assumption that due to the already low levels of MGMT in the bone marrow, the toxicity in the bone marrow would not significantly increase should later be proven wrong [61].

Quinn et al. reported in a phase I trial [62] and in a phase II trial [63], where TMZ plus O^6^-benzylguanine was administered to patients with recurrent, TMZ-resistant glioblastoma, that myelosuppression was most commonly identified. Patients experienced grade 4 neutropenia, grade 4 thrombocytopenia, grade 4 lymphopenia and grade 3 and 4 anemia, which required a TMZ dose reduction in several patients. Although they observed MGMT depletion after O^6^-benzylguanine administration in blood samples [62], they did not observe a TMZ sensitization in MGMT unmethylated patients [63]. Therefore, O^6^-benzylguanine was not included in the standard therapy of GBM patients.

#### 2.4.2. PARP Inhibitors

The poly(ADP-ribose) polymerase (PARP) family consists of 18 PARP enzymes mainly involved in DNA damage repair and programmed cell death. PARP-1 and PARP-2 are activated upon DNA damages caused, for example, by ionizing radiation or alkylating agents to repair the DNA damage via the base-excision repair (BER) pathway [64]. Both, PARP-1 and PARP-2, were found to increase the antitumor effects of cytotoxic agents and offer treatment options for chemo- and radiosensitization.

PARP-1 binds to the damage on the DNA and generates poly ADP-ribose (PAR) polymers using NAD^+^. Further polymers are transferred to histones and chromatin-associated proteins on the DNA [65]. Once the repair enzymes are recruited, PARP-1 is released from the DNA break to give way for XRCC1. XRCC1 assembles the repair enzymes and factors onto the DNA to repair the break. While the DNA is repaired, PARP-1 gets reactivated by the glycohydrolase PARG removing the PARylations [65]. Therefore, PARP-1 enhances cell survival and mediates resistance to radio- and chemotherapy.

PARP inhibitors either inhibit NAD^+^ binding and following PARylation or trap PARP, thereby, blocking the damaged site for repair enzyme assembly [65]. Both lead to replicative stress and DNA double-strand breaks [66].

Several PARP inhibitors are currently tested in phase I, II and III clinical trials, including olaparib, iniparib, pamiparib, niraparib, veliparib, and talazoparib. Dungey et al. showed that olaparib increased radiosensitivity of GBM cells in vitro due to collapsed replication forks after radiation treatment [67]. They propose that the radiosensitizing effect occurs due to the replicating cells necessitating a fractionated treatment regimen [67]. Here, the PARP inhibitor does not directly have an effect on MGMT but rather on DNA replication, making it a good example for the radiosensitizing effects of PARP inhibitors.

Veliparib, in contrast, was found an alternative treatment option for MGMT unmethylated GBM patients as a combination of veliparib with irradiation inhibited cell proliferation in MGMT unmethylated primary cell lines as well as increased survival and apoptosis and decreased cell proliferation in vivo [68]. However, a randomized phase I/II study from 2016 combining TMZ and veliparib in recurrent GBM patients did not significantly increase overall survival and progression-free survival [69]. However, the results of a more recent published phase II trial (2019) comparing standard of care to veliparib concomitant to radio- as well as chemotherapy indicate an advantage of veliparib compared to standard of care treatment with an extended six months progression-free survival [70]

PARP inhibitors present a novel, innovative and personalized treatment option for MGMT unmethylated GBM patients; however, clinical trials are currently ongoing and analyses need to be completed before adding PARP inhibitors to the standard treatment of GBM.

## 3. miRNA

microRNAs (miRNA) are small non-coding RNA molecules consisting of 19-22 nucleotides first described in Caenorhabditis elegans in 1993 [71]. Lee et al. discovered that the *lin-4* gene produces short RNAs that are complementary to the 3′UTR of *lin-14* mRNA and further observed a down-regulation of LIN-14 protein. This led them to the assumption that the direct RNA-RNA interaction between the *lin-4* transcript and the *lin-14* 3′UTR leads to LIN-14 protein down-regulation [71]. Further, they proposed the existence of a class of regulatory genes producing small antisense RNAs influencing gene expression later to be known as microRNAs [71].

In 2001, the word microRNA was first introduced by Lagos-Quintana et al. [72] who could show that many miRNAs are expressed in several species and are highly conserved. The main role of miRNAs is posttranscriptional regulation by sequence-specific repression of mRNAs [72]. To date, more than 2000 miRNAs have been discovered in the human genome [73], which each regulates hundreds of targets including genetic pathways, indicating their role in gene regulation, disease development and also tumorigenesis [74].

### 3.1. miRNA Biogenesis

miRNAs are initially produced in the nucleus from large hairpin looped RNA precursors by the RNA polymerase II [75,76]. These precursors are termed pri-miRNAs and are processed to pre-miRNAs of varying length [75] by the RNase III enzyme Drosha [77,78] and the double-stranded RNA-binding protein Pasha [79]. Via exportin 5 [80], the pre-miRNAs get exported into the cytoplasm [75], where the RNAse III enzyme Dicer processes it to 22 nucleotides long double-stranded RNAs that form the miRNA: miRNA*duplex. The mature miRNA is unwound and released from Dicer [75] and Argonaut protein 2 (Ago2) [81] mediates the assembly to the multiprotein RNA-induced-silencing complex (miRISC) [81]. Which strand eventually enters the miRISC depends on the internal strand stability [82]. The end of the strand with the lowest stability is likely to be the target of a helicase-like enzyme, which unwinds the duplex [82]. Here, the 5′ end exhibits the lowest internal stability. Perfect complementarity between the mature miRNA and the mRNA target leads to the cleavage of the target mRNA, whereas imperfect complementarity only leads to translational repression [83].

Dysregulation of miRNAs due to gene deletions, amplifications and translocations or defects in the miRNA biogenesis machinery seem to be the mechanisms contributing to the malignant cell types eventually leading to cancer.

### 3.2. miRNA in Cancer

In 2002, Calin et al. were the first to discover an association between miRNA dysregulation and cancer: a deletion on chromosome 13q14 coding for the *miR15* and *miR16* genes was observed in more than half of the B-cell chronic lymphocytic leukemia (CLL) and deletions or down-regulations of miR-15 and miR-16 were observed in 68% of the B-cell chronic lymphocytic leukemia [84].

Further, in 2004, they published that miRNAs are either tumor suppressive or oncogenic depending on their location; located at regions of loss of heterozygosity suggests tumor suppressors, while located at regions of amplifications suggests oncogenes [85]. In their genome-wide examination, they discovered an association between miRNA location and cancer. miRNAs are commonly found at cancer-associated regions, in which loss of heterozygosity regions may contain tumor suppressor genes and amplifications harbor oncogenes or the other way around [85]. An example for tumor suppressive miRNAs are miR-15 and miR-16, as their absence due to deletions on chromosome 13 leads to CLL. Further, it was shown by Cimmino et al. that the deletion of miR-15 and miR-16 leads to increased expression of Bcl-2 resulting in the formation of leukemias and lymphomas [86]. From this discovery, they proposed tumor suppressive miRNAs as inhibitors of their oncogenic targets in cancer therapy. Another mechanism for dysregulation of miRNAs in cancer apart from deletions and amplifications is the control of the transcription factors. The dysregulation of transcription factors regulating, for example, cell cycle progression, apoptosis, autophagy, invasion, and neoangiogenesis is tightly linked to cancer development. A key regulator of cell cycle progression and a commonly known tumor suppressor gene is p53. Mutation of p53 is frequently found in many cancers and its interaction with miRNAs suggests tumor suppressive features. Yamakuchi and Lowenstein discovered that miR-34a expression is induced by p53, which in turn suppresses p53, negatively regulating SIRT1 to induce apoptosis [87].

Therefore, miRNAs play an important role in tumor development, progression and recurrence. However, miRNAs also represent an innovative treatment option as prognostic and diagnostic biomarkers as well as therapeutic targets in cancer therapy [11,88,89].

The most common upregulated miRNA in many cancers is miR-21. miR-21 is an oncogenic miRNA inhibiting key regulator of apoptotic genes [90]. It was first found to be significantly upregulated in human glioblastoma and its inhibition leads to increased caspase activation followed by apoptotic cell death [90]. Therefore, miR-21 is an example of an oncogenic miRNA, in which upregulation is associated with cancerogenesis. Various bioinformatics and experimental studies have tried to identify a set of de-regulated miRNAs in glioblastoma that are responsible for this tumor.

### 3.3. miRNA in GBM

The dysregulation of miRNAs in several cancers was shown to contribute to cancer development and progression. These miRNAs, their targets, prognostic and diagnostic value, as well as their potential in the treatment of GBM, need to be identified. Table 1 gives an overview of some miRNAs already discovered in GBM, their targets (if known) and their prognostic value (if available). This table gives a small insight into some of the most important miRNAs in GBM and, by far, does not include all up to date identified miRNA dysregulated in GBM.

### 3.4. miRNAs Targeting MGMT

Since the discovery of the importance of the MGMT promoter methylation status in GBM therapy outcome [39], it is now known that the promoter methylation is not the only deterministic factor for MGMT protein expression. In 2013, Kreth et al. discovered the presence of two MGMT transcripts, which are both expressed in GBM [52]. In normal brain tissue, only the shorter transcript with a size of 440 bp is found, which contains a canonical poly(A) signal as well as a 3′UTR of 105 nt. The longer transcript of about 850 bp contains an alternative poly(A) signal of 522 nt and is found only in GBM. In patient samples, they discovered that the length of the transcript is associated with high or low MGMT expression; high MGMT expression correlated with the normal 3′UTR length, whereas reduced MGMT expression levels were associated with the MGMT transcript containing the elongated 3′UTR [52]. Analysis of potential miRNA binding sites revealed 29 miRNAs specific for the long 3′UTR and only two for the short 3′ UTR; miR-181d was found in both. This led to the conclusion that longer UTRs render transcripts more accessibility to miRNA targets.

First, an in silico analysis of miRNAs targeting MGMT using the TarBase v.8 online tool (DIANA-LAB, Biomedical Sciences Research Center Alexander Fleming, Vari, Greece) [131] was done. The 43 found miRNAs are present in Table 2 below.

Table 3 gives an overview of those miRNAs regulating MGMT expression, which were identified experimentally and which exhibit significant effects in cell lines. Detailed descriptions follow in the sections below.

#### 3.4.1. miR-142-3p

Lee et al. determined an inverse correlation between MGMT and miR-142-3p expression levels in GBM cell lines: high MGMT expressing cell lines show low levels of miR-142-3p and low MGMT expressing cell lines show high levels of miR-142-3p [110]. In miR-142-3p overexpression experiments, no change in MGMT mRNA expression was observed, but a reduction in MGMT protein expression. This leads to the assumption that miR-142-3p directly interacts with the 3′UTR of MGMT, which was further proven by luciferase reporter assay experiments [110]. Additionally, an increased sensitivity towards alkylating agents was determined using TMZ and BCNU in miR-142-3p overexpressing cell lines with a stronger effect when BCNU was added [110].

Previously, the same workgroup reported that miR-142-3p is suppressed by the oncogenic cytokine IL6 promoting GBM propagation, suggesting that miR-142-3p is a tumor suppressive miRNA [111].

Taken together, miR-142-3p regulates MGMT protein expression and sensitizes cells in vitro to alkylating agents, which might indicate a potential biomarker for individual GBM treatment [111].

#### 3.4.2. miR-181d

Zhang et al. were the first to identify a miRNA regulating MGMT. miR-181d post-transcriptionally regulates MGMT by direct interaction with the long 3′UTR MGMT transcript [116,135]. In vitro experiments could show that transfection with miR-181d significantly downregulated MGMT mRNA as well as MGMT protein expression and sensitized cells to TMZ [116]. Further analysis of glioblastoma patient samples indicated that miR-181d is usually down-regulated and that transfection with a mimic in vitro inhibits cell proliferation by targeting K-ras, promotes G1 cell cycle arrest and induces apoptosis by targeting Bcl-2 [115]. Evaluation of clinical data also revealed that a higher miR-181d expression was associated with improved overall survival [116] and that miR-181d expression levels increased after either TMZ or irradiation alone and significantly increased after irradiation and TMZ treatment combined [136]. This suggests that miR-181d could act as a predictive biomarker for chemo- and radiotherapy outcomes.

Several studies have investigated the effect of miR-181d and MGMT expression and discovered similar results to Zhang et al. Interaction between miR-181d and other miRNAs, such as miR-409-3p [133], miR-648 and miR-661 [135], have been found to enhance the effect of MGMT down-regulation, suggesting that miR-181d is the key miRNA regulating MGMT expression.

Taken these factors together miR-181d as a tumor suppressive miRNA could be of great use in treating glioblastoma patients to increase sensitivity to TMZ by directly targeting MGMT mRNA [135]. To the best of our knowledge, miR-181d is the only miRNA that regulates MGMT and is associated with overall survival. Up to date, there are no clinical trials ongoing investigating miR-181d as an innovative treatment option.

#### 3.4.3. miR-221/222

miR-221/222 have been extensively studied in various cancers and were shown to be overexpressed in glioblastoma, prostate carcinoma, papillary thyroid carcinoma, hepatocellular cancer and pancreatic cancer [122]. Gillies and Lorimer demonstrated that miR-221/222 are upregulated in human glioblastoma and target p27, a cell cycle regulator [121]. Further targets include the Akt signaling pathway, PTEN, TIMP-3, as well as MMP-2 and MMP-9 [122,137]. In vitro overexpression of miR-221/222 resulted in the induction of p-Akt, MMP-2, and MMP-9 protein expression and hence increased cell proliferation and invasion. These results were confirmed in in vivo overexpression experiments, which also led to increased tumor growth as well as morphological changes towards a malignant phenotype [122].

A binding site of miR-221/222 was found at the 3′UTR of MGMT and further confirmed in in vitro experiments [123]. Overexpression of miR-221/222 reduced MGMT levels in transfected human glioblastoma cell lines and increased the cells’ sensitivity to TMZ [123]. It can be concluded that miR-221/222 are oncogenic miRNAs negatively influencing patients’ survival, however, increasing sensitivity to TMZ in vitro by directly targeting MGMT.

#### 3.4.4. miR-370-3p

Peng et al. were the first to discover a suppressive potential of miR-370-3p in human glioblastoma [114]. miR-370-3p is significantly down-regulated in low- and high-grade gliomas (Grade II and IV) and also in glioblastoma cell lines. Upon transfection with a miR-370-3p mimic cell viability decreased, long-term proliferation was suppressed as well as the percentage of cells arrested in S and G2/M phase of the cell cycle decreased [114]. A direct post-transcriptional target was found in the 3′UTR of ß-catenin, which is involved in the Wnt signaling pathway promoting cell proliferation and migration [114].

Gao et al. found similar results: in recurrent GBM miR-370-3p expression was significantly decreased compared to normal brain tissue, GBM cell lines showed low levels of miR-370-3p, as well as miR-370-3p-transfected cells showed decreased proliferation [132]. Cell lines expressing the lowest miR-370-3p were more resistant to TMZ compared to cell lines expressing higher levels of miR-370-3p [132]. Additionally, they could demonstrate a negative correlation between MGMT mRNA and miR-370-3p expression.

In 2018, Nadaradjane et al. postulated that miR-370-3p is a biomarker for the prediction of GBM treatment planning and therapy outcome. However, they found out that the expression level of miR-370-3p in the blood of GBM patients varies during standard treatment and is not associated with overall survival [127]. Still, they observed a longer patient survival when miR-370-3p overexpression lasted longer before relapse occurred. In vitro they could show that miR-370-3p overexpression leads to decreased MGMT mRNA and decreased MGMT protein levels. Further, miR-370-3p increased the cells’ sensitivity to TMZ indicated by increased cell death after treatment. Subcutaneous tumors grown in mice and treated with a combination of TMZ and miR-370-3p significantly decreased in volume. In the resected tumors, a significant reduction of MGMT expression was observed [127].

Another target of miR-370-3p is FOXM1, which is involved in cell cycle progression. Upon miR-370-3p overexpression, FOXM1 expression reduced as well [127]. Hence, cell cycle progression was inhibited and cell death induced.

It can be concluded that miR-370-3p is a tumor suppressive miRNA in GBM by downregulating the mRNA and protein expression of MGMT as well as FOXM1 expression. miR-370-3p is not deterministic for patients’ survival but can be used to sensitize to TMZ especially in MGMT unmethylated patients. However, no clinical trials are currently ongoing investigating miR-370-3p as an innovative treatment option.

#### 3.4.5. miR-409-3p

miR-409-3p was found 5-fold upregulated in human GBM samples compared to healthy brain tissue with an inverse correlation between MGMT and miR-409-3p expression [133]. Patient samples with low MGMT show high miR-409-3p levels, while high MGMT expressing samples show low miR-409-3p levels. In vitro transfection with a miR-409-3p mimic of the high MGMT expressing cell line T98G demonstrated a significant down-regulation of MGMT mRNA as well as MGMT protein. This suggests that miR-409-3p is a strong inhibitor of MGMT by the degradation of MGMT mRNA as well as by translational repression [133]. An even more enhanced effect of MGMT suppression was observed when miR-409-3p mimics were cotransfected with miR-181d mimics [133].

As miR-409-3p was found significantly downregulated in human GBM samples repressing MGMT expression, it can be concluded that miR-409-3p might be a potential therapeutic approach to sensitize MGMT unmethylated patients to alkylating chemotherapeutics. However, other targets of miR-409-3p are still unknown, but Khalil et al. suggested a possible protective role in pro-angiogenic and pro-metastatic processes [133].

#### 3.4.6. miR-603

miR-603 is found upregulated in glioblastoma samples and promotes cell proliferation as well as cell cycle progression [130]. Targets of miR-603 include WIF1 and CTNNBIP1 activating the Wnt/ß-catenin signaling pathway and promoting cell proliferation and migration [130]. Therefore, miR-603 can be considered an oncogenic miRNA.

MGMT is directly suppressed by the interaction of miR-603 with the 3′UTR of MGMT [134]. Transfection with a miR-603 mimic significantly reduced MGMT mRNA levels and protein expression and further enhanced sensitivity to TMZ in vitro as well as in vivo [134]. Kushwaha et al. also showed that the combination of miR-181d and miR-603 most effectively regulated MGMT expression compared to either alone [134].

For innovative treatment options, miR-603 might be a promising candidate to inhibit MGMT and Wnt/ß-catenin signaling pathway activation. No clinical trials have been proposed yet.

#### 3.4.7. miR-648, miR-661 and miR-767-3p

When Kreth et al. discovered the presence of two MGMT isoforms either containing a long or a short UTR, they used target prediction software to determine miRNA with a binding site within the UTRs of MGMT [52]. They assumed that these miRNAs are expressed in human GBM and negatively correlate with MGMT expression. In human GBM samples, six miRNAs (miR-184, miR-183, miR-661, miR-370, miR-767-3p, and miR-648) were found binding exclusively in the long UTR, two (miR-1197 and miR-655) within the short UTR and one (miR-181-d) in both UTRs [135]. Upon cloning both UTRs into a reporter vector containing two luciferases and co-transfection with the miRNAs, they observed that the short UTR-binding miRNAs (miR-181d, miR-665, and miR-1197) did not show regulatory activity [135]. This indicates that the short UTR of MGMT is not regulated by these miRNAs. Five (miR-661, miR-370, miR-181d, miR-767-39, and miR-648) of the seven miRNAs possibly regulating the long UTR showed significant luciferase repression but only three miRNAs (miR-181d, miR-767-3p and miR-648) showed decreased MGMT protein expression. Here, miR-648 exerted the strongest MGMT protein reduction. In qPCR experiments, only two miRNAs (miR-181d and miR-767-3p) significantly reduced MGMT mRNA expression, indicating that those two regulate MGMT expression via direct degradation of the mRNA transcript and miR-648 might act via translational repression [135].

Further, they could show that miR-767-3p and miR-648 are significantly upregulated in human GBM samples and that cotransfection with all three miRNAs (miR-181d, miR-767-3p and miR-648) significantly increased the sensitivity to TMZ treatment [135]. These data correlate with data from Jesionek-Kupnicka et al., who also found an association between MGMT and miR-181d and miR-648 expression [138].

### 3.5. miRNAs as Innovative Treatment Option for GBM

To the best of our knowledge, no clinical trials are currently ongoing investigating the above-mentioned miRNAs as innovative treatment options for GBM patients nor have any miRNA-based therapies been approved by the FDA. Target specificity and tissue toxicities are major problems in the delivery of miRNA or miRNA inhibitors to their mRNA target.

Several invasive strategies have been postulated to enhance drug delivery across the BBB including intracerebral implants, disruption of the BBB, intra-arterial and intrathecal drug delivery, direct injections into the brain, catheters, pumps or microdialysis [139]. As all of these strategies require invasion into the brain tissue or tumor tissue, there is an increased risk for brain damage and other side effects, including toxicities, indicating the urgent need for non-invasive strategies. Therefore, biological strategies have been developed as innovative tools for drug delivery. These strategies include RNA interference, viral vectors, exosomes, antisense therapy, gene therapy, antibody conjugates, peptide carriers and other carriers [139,140]. Also, chemical systems have been developed, such as lipophilic analogues, prodrugs, efflux transporter inhibition, liposomes, nanoparticles, polymeric micelles and dendrimers [139,140]. Both biological and chemical strategies allow for target specific delivery, as it is most important and challenging at the same time to deliver and internalize the drug or miRNA specifically to the tumor. The challenges of designing nanoparticles are reviewed elsewhere [141].

However, the most limiting factor in delivering these compounds into the brain and promoting restricted bioavailability is the BBB. Major issues are the enzymatic degradation of the miRNA or miRNA inhibitors themselves before the target can be reached as well as the inability of packaging molecules due to high molecular weight and polar functional groups [139]. The BBB is a natural barrier against toxins, harmful substances, and fluctuations in chemical concentrations [139]. It consists of endothelial cells, forming the walls of the capillaries and epithelial cells, creating the blood-cerebrospinal fluid barrier (BCSFB) [142]. The cerebrospinal fluid is secreted into the brain, while the interstitial fluid is secreted by the capillary endothelium [142]. These two fluids can communicate in order to regulate fluctuations and maintain a stable environment [142]. The avascular arachnoid epithelium is the enclosing layer sealing the extracellular fluids from the rest of the body [142]. Physical barriers such as tight junctions, transport barriers such as transporters, and metabolic barriers including enzymes, are found at all interfaces representing the protecting characteristics of the BBB. The most important factor thereby are tight junctions, significantly reducing the trespassing of polar solutes by blocking their penetration [142]. The only routes molecules and solutes can penetrate the BBB are via passive diffusion and ABC transporter efflux (lipid, soluble, non-polar molecules), via solute carriers (e.g., glucose, amino acids, small peptides), via transcytosis or receptor- and adsorptive-mediated (e.g., lipoproteins, insulin, glycosylated proteins, histones) or leukocytes via diapedesis [142].

However, all these strategies need further characterization, experimentation, and clinical trials to safely deliver molecules, miRNAs and other compounds to specific target sites. Up to date, only some miRNAs, including miR-122, -21, -155, -92 and -29 are currently tested in clinical trials as targeted therapy for Hepatitis C (HCV), nephritis, CLL, wounds and fibrosis [143,144]. Only two miRNAs are currently tested for the use in cancer therapy: a miR-16 mimic is involved in a Phase I trial for non-small-cell lung cancer [144] and another clinical trial testing a miR-34a mimic for hepatocellular carcinoma (HCC) has recently been terminated [145]. MRX34, a synthetic, 23 nt long double-stranded RNA encapsulated in a liposomal nanoparticle was administered to patients mainly suffering from HCC. Although pre-clinical studies in non-human primates showed promising results, severe adverse effects and also death of four patients due to the drug forced the phase I trial to be terminated [145]. Severe adverse effects were unlikely due to the liposomal carrier, but rather due to severe immune-related toxicities, which have yet to be resolved [145].

## 4. Conclusions

In the last decade, miRNAs have become promising tools as prognostic and diagnostic biomarkers as well as therapeutic targets for innovative and personalized cancer treatment [11,89,90]. Several miRNAs have been found differentially expressed and predictive for overall survival, progression-free survival or treatment outcome in several cancer entities. Some miRNAs such as miR-21, the miR-17 cluster and miR-221/222 are dysregulated in several cancer types, but most importantly, cancer type-specific miRNA signatures were also discovered [89,146,147,148,149].

With a survival rate of less than 3% [3], Glioblastoma multiforme presents an urgent need for new innovative and personalized treatment options. Patients with a wildtype IDH and an unmethylated MGMT promoter region have the poorest prognosis and the shortest survival [150], identifying these patients with the most urgent need for new treatment options. In this review, we focused on MGMT unmethylated patients and tried to identify possible miRNAs regulating MGMT expression, which could be used for personalized treatment in the future.

We identified eight promising miRNAs—miR-142-3p, -181d, -221/222, -370-3p, -409-3p, -603, -648, and -767-3p—negatively regulating MGMT expression either via mRNA degradation or translational repression. Five of these miRNAs (miR-142-3p, -181d, -221/222, -370-3p and -603) were positively tested to increase sensitivity to alkylating agents such as BCNU and TMZ in vitro as well as in vivo [110]. miR-181d was the only miRNA found predictive for overall survival [116,136].

We present here miRNAs that could help reduce and repress MGMT expression by targeted treatment to sensitize the tumors against alkylating agents. However, target-specific delivery, especially into the brain, represents a challenging task, which has yet to be overcome.

## Figures and Tables

**Table 1 cancers-12-01099-t001:** miRNAs in Glioblastoma multiforme, their targets, function, and prognostic value (↓ = decreased, ↑ = increased)

microRNA	Regulation	Type	Target	Function	Prognosis	Ref.
miR-10b	up	oncogenic	uPAR, RhoC	↑invasion		[91]
miR-7	down	tumor suppressor	EGFR	↑apoptosis, ↓cell proliferation, ↓migration, ↓invasion		[92,93]
miR-17	up	oncogenic	DFFA, PI2KCA, E2F3m VEGFA, ATG7	↑autophagy		[94,95]
miR-21	up	oncogenic	HNRPK, TAp63, PTEN, EGFR, E2F3, PDCD4, WNT5A	↓apoptosis, ↓autophagy, ↑invasion		[88,93,94,95,96]
miR-26a	up	oncogenic	PTEN	↑tumor growth ↑angiogenesis	high level = longer OS and PFS with carmustine ↑TMZ resistance	[97,98,99]
miR-34a	down	tumor suppressor	E2F3, PI2KCA, EGFR, DFFA, CSL2, BAX, c-Met, Notch	↑cell cycle arrest, ↓invasion ↓migration ↓cell proliferation		[95,100]
miR-92b-3p	up	oncogenic	PTEN	↑migration, ↑invasion ↓apoptosis	low level = shorter OS	[101,102]
miR-124	down	tumor suppressor	CDK6	↓cell cycle progression		[103]
miR-125b	up	oncogenic	p53, p38MAPK, Bmf	↑proliferation, ↑cell cycle progression,	high level = higher grade	[104,105]
miR-128	down	tumor suppressor	RTKs, EGFR, PDGF-R, E2F3a	↓proliferation, ↑differentiation, ↓migration		[93,106,107]
miR-130a	up	tumor suppressor	E2F8	ROS production	high level = extended survival without progression predictor for TMZ response	[108,109]
miR-137	down	tumor suppressor	CDK6	↓cell cycle progression		[103]
miR-142-3p	down	tumor suppressor	IL-6, HMGA2	↓cell viability	high levels = low MGMT low levels = high MGMT	[110,111]
miR-155	up	oncogenic	FOXO3a	↑proliferation, ↑migration, ↑invasion	low level = long OS	[108,112]
miR-181a	down	tumor suppressor	Bcl-2	↑apoptosis		[113]
miR-181b	down	tumor suppressor	SALL4	↓proliferation, ↓migration, ↓invasion		[114]
miR-181d	down	tumor suppressor	MGMT, Bcl-2, KRAS	↓proliferation, ↓cell cycle progression, ↑apoptosis	high level = improved OS	[115,116]
miR-210	up	oncogenic	SIN3A	↑proliferation, ↓apoptosis	low level = long OS	[108,117]
miR-218	down	tumor suppressor	IKK-ß, LEF1, Bmi1	↓invasion, ↓migration, ↓proliferation, ↑apoptosis		[93,118,119,120]
miR-221/222	up	oncogenic	p27, AKT, TIMP-3, PTEN, E2F3	↑proliferation, ↑invasion	up in short-term, down in long-term survivors, ↑TMZ sensitivity ↓long-term survival	[121,122,123]
miR-326	down	tumor suppressor	WNT5A, TOM34		high level = extended survival without progression	[95,108]
miR-335	up	oncogenic	DAAM1, PAX6	↑proliferation, ↑invasion		[124,125]
miR-339	up	oncogenic		↑migration, ↑invasion ↓apoptosis		[101]
miR-370-3p	down and up	tumor suppressor	ß-catenin, FOXM1	↓cell proliferation ↓cell cycle progression	upregulation = inhibition of GBM growth long upregulation = longer survival	[126,127]
miR-409	down	tumor suppressor	HMGN5, cyclin D1, MMP2	↑invasion, ↑proliferation		[128]
miR-451	down	tumor suppressor	Cyclin D1, p27, Bcl-2, MMP-2, MMP-9	↓cell cycle ↓cell growth ↑apoptosis ↓cell growth		[129]
miR-603	up	oncogenic	WIF1, CTNNBIP1	↑proliferation, ↑cell cycle progression		[130]

**Table 2 cancers-12-01099-t002:** miRNAs targeting O^6^-methylguanine–DNA methyltransferase (MGMT) derived from an in silico analysis.

microRNAs	
let-7a-2-3p	miR-342-3p
let-7f-2-3p	miR-361-5p
let-7i-3p	miR-3619-5p
miR-1-3p	miR-370-3p
miR-16-5p	miR-371a-3p
miR-17-5p	miR-374a-5p
miR-20a-3p	miR-379-5p
miR-21-3p	miR-423-3p
miR-27a-3p	miR-429
miR-27a-5p	miR-497-3p
miR-30d-5p	miR-548a-3p
miR-30e-3p	miR-561-3p
miR-155-5p	miR-589-3p
miR-181b-5p	miR-603
miR-181d-5p	miR-612
miR-183-5p	miR-616-3p
miR-184	miR-648
miR-191-5p	miR-651-5p
miR-194-5p	miR-661
miR-324-5p	miR-767-3p
miR-325	miR-2115-5p
miR-338-5p	

**Table 3 cancers-12-01099-t003:** miRNAs involved in MGMT regulation in Glioblastoma multiforme.

microRNA	Regulation	Type	Prognosis	Ref.
miR-142-3p	down	tumor suppressor	suppression of MGMT protein ↑TMZ sensitivity	[110,111]
miR-181d	down	tumor suppressor	degradation of MGMT mRNA; high level = improved OS	[116]
miR-221/222	up	oncogenic	suppression of MGMT; ↑TMZ sensitivity	[123]
miR-370-3p	down and up	tumor suppressor	regulatory effects on MGMT; ↑TMZ sensitivity	[127,132]
miR-409-3p	up	oncogenic	repression of MGMT	[133]
miR-603	up	oncogenic	suppression of MGMT ↑TMZ sensitivity	[134]
miR-648	up	tumor suppressor	inhibition of MGMT protein translation	[135]
miR-767-3p	up	tumor suppressor	degradation of MGMT mRNA	[135]

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
