# Peer review of "The Role of miRNA for the Treatment of MGMT Unmethylated Glioblastoma Multiforme"

_cancers, 2020, doi:10.3390/cancers12051099_

Round 1

Reviewer 1 Report

To authors,

Kirstein et al’s review manuscript entitled “The role of miRNA for the treatment of MGMT unmethylated Glioblastoma multiforme” claimed that they reviewed some miRNAs in glioblastoma and their implications in TMZ resistance. Overall, manuscript covers their main subject well enough to provide readers about up-to-date information on miRNAs and their relationship in TMZ resistance. Description about overall pathological information of glioma and chemical/pharmacological of MGMT are well written and deep enough to understand their histological background. The manuscript will benefit readers who are interesting in the field. Major and minor comments are listed below.

Major comments:

  1. Throughout the manuscript, there are many descriptions not citing original publications. For examples in page 10, “miR-221/222 have been extensively studied in various cancers and were shown to be overexpressed in glioblastoma, prostate carcinoma, papillary thyroid carcinoma, hepatocellular cancer and pancreatic cancer [111]. Gillies and Lorimer demonstrated that miR-221/222 are upregulated in human glioblastoma and target p27, a cell cycle regulator [110]. Further targets include the Akt signaling pathway, PTEN, TIMP-3, as well as MMP-2 and MMP-9 [111]. In vitro overexpression of miR-221/222 resulted in the induction of p-Akt, MMP-2, and MMP-9 protein expression and hence increased cell proliferation and invasion. These results were confirmed in in vivo overexpression experiments, which also led to increased tumor growth as well as morphological changes towards a malignant phenotype [111].” Repeatedly cited Ref 111. 111 is not a review paper describing all. However, PTEN/TIMP-3 downregulation by miR-221/222 was reported by Garofalo et al in 2009 in Cancer Cell. Accuracy and originality in citation is a critical factor to appeal a review paper as informative to the readers.

Minor comments:

  1. Line 122 on page 3, “A 4-miRNA signature consisting of let-7b-5p, miR-125a-5p, miR-615-5p and let-70-5p was proposed to assign patients into high- and low-risk groups [31]” seems not right. The Ref 31 reports “miRNAs hsa-let-7a-5p, hsa-let-7b-5p and hsa-miR-125a-5p in patients of the higher-risk group show a tendency towards lower expression and that of hsa-miR-615-5p a tendency towards higher expression”. Please, correct the description accordingly. Typo “let-70-b” should be corrected to let-7a-5.
  2. Line 122-125 on page 3, the description “All four miRNAs are tumor suppressive in GBM or other cancer entities and are higher expressed in the low-risk GBM group [31]. This leads to the promising conclusion, that a high expression of this 4-miRNA signature is associated with a lower malignancy of GBM” is wrong. The Ref 31 clearly stated that “Expressions of miRNAs hsa-let-7a-5p, hsa-let-7b-5p and hsa-miR-125a-5p positively correlated with overall survival and that of hsa-miR-615-5p negatively correlated with overall survival”. Those 4 miRs show significant associate with patient survival despite their role. The paragraph needs to be rewritten.
  3. Line 165 on page 4, it is not clear what “So, the absence or presence mainly contributes” indicates. Of MGMT, or of methylation on MGMT’s promotor region.
  4. Line 255 on page 6, pre-miRNAs’ length are vary, not 70 for all.
  5. Line 273 on page 6, “they published that miRNAs are either tumor suppressive or oncogenic depending on their target gene [77]” is misleading. As Calin et al in the Ref 77 proposed, oncogenic vs TS of miRNAs are depending on their expression levels as found in deletion/fragile sites or amplification sites.
  6. Line 489 on page 12, “Only one miRNA is currently tested for the use in cancer therapy: a miR-16 mimic is involved in a Phase I trial for non-small-cell lung cancer [129].” To emphasize the difficulty of miRNA-based cancer therapy, it’d be helpful for readers to provide the information about miR-34a delivery to HCC patients in phase I trial which has terminated in 2018 after serious sideeffects in patients.

Author Response

1st Reviewer

Major comments:

Throughout the manuscript, there are many descriptions not citing original publications. For examples in page 10, “miR-221/222 have been extensively studied in various cancers and were shown to be overexpressed in glioblastoma, prostate carcinoma, papillary thyroid carcinoma, hepatocellular cancer and pancreatic cancer [111]. Gillies and Lorimer demonstrated that miR-221/222 are upregulated in human glioblastoma and target p27, a cell cycle regulator [110]. Further targets include the Akt signaling pathway, PTEN, TIMP-3, as well as MMP-2 and MMP-9 [111]. In vitro overexpression of miR-221/222 resulted in the induction of p-Akt, MMP-2, and MMP-9 protein expression and hence increased cell proliferation and invasion. These results were confirmed in in vivo overexpression experiments, which also led to increased tumor growth as well as morphological changes towards a malignant phenotype [111].” Repeatedly cited Ref 111. 111 is not a review paper describing all. However, PTEN/TIMP-3 downregulation by miR-221/222 was reported by Garofalo et al in 2009 in Cancer Cell. Accuracy and originality in citation is a critical factor to appeal a review paper as informative to the readers.

  • We thank the Reviewer for this valuable comment. We added Garofalo et al in 2009 in Cancer Cell as suggested to the references.

Minor comments:

  1. Line 122 on page 3, “A 4-miRNA signature consisting of let-7b-5p, miR-125a-5p, miR-615-5p and let-70-5p was proposed to assign patients into high- and low-risk groups [31]” seems not right. The Ref 31 reports “miRNAs hsa-let-7a-5p, hsa-let-7b-5p and hsa-miR-125a-5p in patients of the higher-risk group show a tendency towards lower expression and that of hsa-miR-615-5p a tendency towards higher expression”. Please, correct the description accordingly. Typo “let-70-b” should be corrected to let-7a-5.
  • The reference contains the following statement:

“We could extract a 4-miRNA signature which, with high statistical significance, allowed differentiating between high- and low-risk GBM patients independently of the MGMT methylation status” taken from the original literature page 45770 (10.18632/oncotarget.9945).

  1. Line 122-125 on page 3, the description “All four miRNAs are tumor suppressive in GBM or other cancer entities and are higher expressed in the low-risk GBM group [31]. This leads to the promising conclusion, that a high expression of this 4-miRNA signature is associated with a lower malignancy of GBM” is wrong. The Ref 31 clearly stated that “Expressions of miRNAs hsa-let-7a-5p, hsa-let-7b-5p and hsa-miR-125a-5p positively correlated with overall survival and that of hsa-miR-615-5p negatively correlated with overall survival”. Those 4 miRs show significant associate with patient survival despite their role. The paragraph needs to be rewritten.
  • The paragraph was rewritten: Three of the four miRNAs – let-7b-5p, let-7a-5p and miR-125a-5p - are tumor suppressive in GBM and are higher expressed in the low-risk GBM group [1]. Only miR-615-5p does not show a tendency towards a certain expression level in either risk group [1]. This leads to the promising conclusion, that this 4-miRNA signature is associated with overall survival of GBM patients.

  1. Line 165 on page 4, it is not clear what “So, the absence or presence mainly contributes” indicates. Of MGMT, or of methylation on MGMT’s promotor region.
  • Here, the absence or presence of MGMT itself is meant. The sentence was changed as follows: So, the absence or presence of MGMT mainly contributes to the chemoresistant character of GBM [2,3].

  1. Line 255 on page 6, pre-miRNAs’ length are vary, not 70 for all.
  • 70 nt was deleted and the sentence changed to:

These precursors are termed pri-miRNAs and are processed to pre-miRNAs of varying length by the RNase III enzyme Drosha and the double-stranded RNA-binding protein Pasha.

  1. Line 273 on page 6, “they published that miRNAs are either tumor suppressive or oncogenic depending on their target gene [77]” is misleading. As Calin et al in the Ref 77 proposed, oncogenic vs TS of miRNAs are depending on their expression levels as found in deletion/fragile sites or amplification sites.
  • We have changed the misleading sentence to:

Further, in 2004, they published that miRNAs are either tumor suppressive or oncogenic depending on their location; located at regions of loss of heterozygosity suggests tumor suppressors, while located at regions of amplifications suggests oncogenes.

  1. Line 489 on page 12, “Only one miRNA is currently tested for the use in cancer therapy: a miR-16 mimic is involved in a Phase I trial for non-small-cell lung cancer [129].” To emphasize the difficulty of miRNA-based cancer therapy, it’d be helpful for readers to provide the information about miR-34a delivery to HCC patients in phase I trial which has terminated in 2018 after serious sideeffects in patients.
  • The following sentence was changed: Only two miRNAs are currently tested for the use in cancer therapy: a miR-16 mimic is involved in a Phase I trial for non-small-cell lung cancer [4] and another clinical trial testing a miR-34a mimic for hepatocellular carcinoma (HCC) has recently been terminated [5].
  • The following sentence was supplemented: MRX34, a synthetic, 23 nt long double-stranded RNA encapsulated in a liposomal nanoparticle was administered to patients mainly suffering from HCC. Although pre-clinical studies in non-human primates showed promising results, severe adverse effects and also death of four patients due to the drug forced the phase I trial to be terminated [5]. Severe adverse effects were unlikely due to the liposomal carrier, but rather due to severe immune-related toxicities, which have yet to be resolved [5].

Reviewer 2 Report

This is an interesting overview of microRNAs that are thought to have interaction with MGMT, a known resistance mechanism in the malignant brain tumour glioblastoma. In general I think the review manuscript is of interest and offers a novel aspect to the concepts of temozolomide (TMZ) resistance. There is one of the clearest and most detailed descriptions of the action of TMZ and the interplay with MGMT that I have read.

I do have some comments and sections that could be improved:

Line 38 - '12th most cause' should be '12th most frequent cause'

'most common malignancy of CNS' should be 'most common primary malignancy of CNS'

The authors should make clear that MGMT methylation status is an independent prognostic factor regardless of treatment (as per Hegi NEJM 2005). Although its most significant effect is through response to TMZ, there must be other mechanisms e.g. possibly mitigating the effect of radiotherapy.

Line 54 - 'circulating microRNAs are extensively studied' should be 'circulating microRNAs have been extensively studied'

59 - proposes should be changed to represents

Line 65 onwards - the classification of GBM has been oversimplified. Other diagnostic sub-categories are recognised in 2017 WHO including histone mutated GBMs. These categories should be made clearer and some description included of the genome wide DNA methylation that has been widely adopted (though I accept not officially part of WHO classifier).

Line 90 - CD133 - this is not universally accepted now as a unique marker of stemness, if included reference should be made to the complexity of this area and the other stem like cell markers that are also recognised

Line 99 - safe resection - other licences techniques and therapies include 5ALA and Gliadel wafers.

114- a should be 'at'

126 - 'all possibilities should be evaluated' - although this is true it remains also true that the only proven treatment in phase 3 trials is the Stupp regime. Sadly, although we recognise it is less likely to be effective in unmethylated patients, there is no current proven alternative and pragmatically it will still be given.

137 - remove 'an'

138 - brain pH varies and may be substantially lower in the hypoxic and acidic tumour microenvironment

175 and 181 - The authors allude to the possible effects of TMZ in unmethylated GBM and the fact that a small number of unmethylated patients still show a response (p=0.06 i.e. nearly significant in the Hegi paper). This should be made more explicit as it is clear other mechanisms are in play as they describe.

Therapies - Why do the authors only describe O6benzylguanine and PARPi as alternatives? Other pathways have been examined and perhaps these could be included. PARPi in particular have many other effects, not directly on MGMT, as do other therapeutic candidates effecting DNA repair e.g. radio sensitisation.

Their list of microRNAs known to effect GBM is extensive but I would also include miR451 as this has been shown to have interesting metabolic effects in GBM.

319 - accessibility change to accessible

375 - grade 2 glioma is not a high grade glioma

The authors should recognise more explicitly the potential for many of their candidate MGMT targeting microRNAs to exert influence via other pathways. Each microRNA will potentially target hundreds of genes and it is difficult to show a specific effect via an individual gene, particularly as in some of the studies multiple microRNAs were transfected. How do they know effects were specifically through MGMT?

A list of microRNAs predicted in silico to target MGMT by targeting prediction softwares would be helpful.

The authors somewhat gloss over the difficulties of using microRNA as therapy. Although many of these targets are promising their use is curtailed by current difficulties in delivering RNA to the correct intracellular location. Surgical delivery strategies or BBB penetrating techniques may get RNA to the CNS but the challenge of internalising RNA to tumour cells before degradation remains. More consideration should given to current mechanisms and limitations e.g. viral delivery, nanoparticle delivery. 

There is significant current interest in circulating microRNAs in blood or CSF as biomarkers for prognosis or treatment response. Do any of the candidate microRNAs the authors describe have measurable levels in blood or CSF?

Reviewer 3 Report

Kirstein A and colleagues presented an interesting review article on the role of MGMT modulation mediated by miRNAs. In particular, the authors first describe the importance of MGMT methylation status in the GBM patients’ responsiveness to Temozolomide-based chemotherapy and subsequently described the epigenetic modulation of MGMT gene mediated by microRNAs (miRNAs). The authors widely described the current knowledge about MGMT modulation mediated by miRNAs, however, the description of the miRNAs involved in GBM should be improved and updated. Overall, the manuscript is well written, however, there are some missing information and some paragraphs are off-topic. Below are reported some minor revisions that will improve the quality of the manuscript:

1) In the chapter “3.2. miRNA in cancer”, the authors start the description of GBM-related miRNAs without a link with the previous paragraph. I suggest combining these two parts better, as follows “Various bioinformatics and experimental studies that have tried to identify a set of de-regulated miRNAs in glioblastoma and responsible for this tumor. One of the most representative miRNAs in GBM is mir-21, found up-regulated in several human tumors, including glioblastoma multiforme…..”. For this purpose, see:
- 10.3892/or.2019.7215
- 10.1371/journal.pone.0188090
- 10.1007/s11060-013-1155-x

2) Provide references for the sentence in line 296-297;

3) Some of the references provided in table 1 date 15-10 years ago when technologies were not accurate and sensitive. Recently, an integrated bioinformatics analysis identified a set of miRNAs significantly de-regulated in GBM thus confirming or denying the results obtained in previous studies reported in this review article. Please integrate the table 1 with these recent findings:
- 10.3892/or.2019.7215
- 10.1371/journal.pone.0188090

4) Please provide references for the following sentences “In the last decade, miRNAs have become promising tools as prognostic and diagnostic biomarkers as well as therapeutic targets for innovative and personalized cancer treatment. Several miRNAs have been found differentially expressed and predictive for overall survival, progression-free survival or treatment outcome in several cancer entities.
Some miRNAs such as miR-21, the miR-17 cluster and miR-221/222 are dysregulated in several cancer types, but most importantly, also cancer type-specific miRNA signatures were discovered.”
For this purpose, see:
- 10.1038/bjc.2013.483
- 10.3892/mmr.2019.9949
- 10.1016/j.omtn.2020.03.003
- 10.3389/fonc.2019.01404

5) I suggest to briefly describe the whole chapter “2.2. Innovative treatment options for MGMT unmethylated patients” as this chapter seems to be a little discordant with respect to the main theme of the review. Therefore, I suggest removing or describing more concisely this part; while, a brief paragraph describing the current diagnostic and prognostic strategies for GBM patients should be added at the end of chapter 2. For this purpose, see:
1) 10.3390/cells8080863
2) 10.1016/j.clineuro.2019.105652
3) 10.1016/j.ebiom.2018.10.024
4) 10.2174/0929867324666170516123206

Author Response

3rd Reviewer

  • In the chapter “3.2. miRNA in cancer”, the authors start the description of GBM-related miRNAs without a link with the previous paragraph. I suggest combining these two parts better, as follows “Various bioinformatics and experimental studies that have tried to identify a set of de-regulated miRNAs in glioblastoma and responsible for this tumor. One of the most representative miRNAs in GBM is mir-21, found up-regulated in several human tumors, including glioblastoma multiforme…..”. For this purpose, see:
    - 10.3892/or.2019.7215
    - 10.1371/journal.pone.0188090
    - 10.1007/s11060-013-1155-x
  • We have slightly changed the order of paragraph 3.2 and added the sentence suggested by the Reviewer in order to create a better transition (Line 353-354):

Various bioinformatics and experimental studies have tried to identify a set of de-regulated miRNAs in glioblastoma that are responsible for this tumor.

  • Provide references for the sentence in line 296-297;
  • We have added the following literatures: 10.1016/j.omtn.2020.03.003, 10.1038/nrc1840 and 10.1158/0008-5472.CAN-06-0800 as references.

  • Some of the references provided in table 1 date 15-10 years ago when technologies were not accurate and sensitive. Recently, an integrated bioinformatics analysis identified a set of miRNAs significantly de-regulated in GBM thus confirming or denying the results obtained in previous studies reported in this review article. Please integrate the table 1 with these recent findings:
    - 10.3892/or.2019.7215
  • The Reference was added to Table 1 for the following miRNAs: miR-7, -21, -128 and -218

  • Please provide references for the following sentences “In the last decade, miRNAs have become promising tools as prognostic and diagnostic biomarkers as well as therapeutic targets for innovative and personalized cancer treatment. Several miRNAs have been found differentially expressed and predictive for overall survival, progression-free survival or treatment outcome in several cancer entities.
  • 1016/j.omtn.2020.03.003, 10.1038/nrc1840 and 10.1158/0008-5472.CAN-06-0800 were added as references
    Some miRNAs such as miR-21, the miR-17 cluster and miR-221/222 are dysregulated in several cancer types, but most importantly, also cancer type-specific miRNA signatures were discovered.”
    For this purpose, see:
    - 10.1038/bjc.2013.483
    - 10.3892/mmr.2019.9949
    - 10.1016/j.omtn.2020.03.003
    - 10.3389/fonc.2019.01404
  • References were added as suggested at appropriate positions in the text and one additional literature: 1371/journal.pone.0188090

  • I suggest to briefly describe the whole chapter “2.2. Innovative treatment options for MGMT unmethylated patients” as this chapter seems to be a little discordant with respect to the main theme of the review. Therefore, I suggest removing or describing more concisely this part; while, a brief paragraph describing the current diagnostic and prognostic strategies for GBM patients should be added at the end of chapter 2. For this purpose, see:
    1) 10.3390/cells8080863
    2) 10.1016/j.clineuro.2019.105652
    3) 10.1016/j.ebiom.2018.10.024
    4) 10.2174/0929867324666170516123206
  • We thank the Reviewer for his comment. We have decided to keep the paragraph “Innovative treatment options for MGMT unmethylated patients” and we have added the following paragraph:

2.3. Current diagnostic and prognostic biomarkers for GBM

The most commonly analyzed biomarkers in GBM are currently IDH status, MGMT status, 1p/19q co-deletion and ATRX loss. There are however several classes of molecules, proposed to aim as biomarkers for GBM detection, which are found in the blood, cerebrospinal fluid (CSF) and urine.

Proteins are detectable in all kinds of body fluids and can be easily withdrawn from the patient. GBM-specific protein markers include VEGF, angiogenesis-associated proteins, extracellular matrix proteins, matrix metalloproteinases, cell line associated proteins, macrophage migration inhibitory factor (MIF) as well as functionally-related proteins, such as CD44. CD44 was shown as a potential marker for survival outcome and treatment resistance. All these have shown deviating amounts an compositions in patients where tumor progression was observed.

Another class used for biomarkers are small molecules, such as lipids and metabolites. Due to their low specificity and small size they can only be used to verify a diagnosis after other markers were tested positive.

Circulating tumor cells (CTCs), which are primary tumor cells circulating in the body via the blood stream e.g. might be important in other cancers apart from GBM. As GBM rarely metastasizes and is described as a cranial-restricted tumor, CTCs might not be found in GBM patients in blood samples.

Extracellular vesicles, secreted by the tumor and containing material characteristic of the parental cells, can be found in the serum as well as the CSF. It is know, that GBM secrete exosomes, microvesicles, apoptotic bodies and oncosomes containing the glioma-specific receptor of epidermal growth factor (EGFRvIII), miR-21 as well as mutant IDH1 mRNA.

Circulating miRNAs have recently gained attention in research and present promising new biomarkers. They can usually be found in peripheral blood of GBM patients and plasma levels of some miRNAs were already shown to be altered. Some of these circulating miRNAs seem predictive in early diagnosis and helpful during treatment monitoring.